# Improving the Assessment Process of Family Functioning in Adult Bipolar Disorders: A PRISMA Systematic Review

**DOI:** 10.3390/jcm11030841

**Published:** 2022-02-05

**Authors:** Caroline Munuera, Philippe Compagnone, Mathilde M. Husky, Paul Lebourleux, Fanny Petit, Katia M’bailara

**Affiliations:** 1LabPsy, University Bordeaux, EA 4139, 33000 Bordeaux, France; caroline.munuera@u-bordeaux.fr (C.M.); philippe.compagnone@u-bordeaux.fr (P.C.); mathilde.husky@u-bordeaux.fr (M.M.H.); fanny.petit@u-bordeaux.fr (F.P.); 2Centre Ressource Bipolaire Sud Aquitain (CReBSA), Clinique Château Caradoc, 24 Avenue du 14 Avril 1814, 64100 Bayonne, France; p.lebourleux@caradoc.fr; 3Centre Hospitalier Charles Perrens, Service de Psychiatrie Adulte, Pôle 3-4-7, 121 rue de la Béchade, 33000 Bordeaux, France; 4Réseau des Centres Expert des Troubles Bipolaires, Département de Psychiatrie, Fondation FondaMental, 40 rue de Mesly, 94000 Créteil, France

**Keywords:** family functioning, bipolar disorders, systematic review, PRISMA, assessment

## Abstract

In order to determine family functioning in the treatment of adults with bipolar disorders, guidelines are needed regarding the way family functioning may be assessed. The present systematic review aims to investigate how family functioning is assessed in this context. Following PRISMA guidelines, a total of 29 studies were reviewed. Results showed that although there was no consensual family functioning assessment across studies, 27 studies (93%) relied on self-report questionnaires, 12 studies (41%) relied on one family member as an informant (adult with bipolar disorder or other) and the adult considered was mostly a woman in the acute phase of bipolar I disorder. Significant heterogeneity was observed in the assessment of family functioning. Methodological considerations regarding the assessment of family functioning are discussed.

## 1. Introduction

Bipolar disorders (BD) are serious mental disorders characterized by chronic alternation between manic and depressive states, separated by euthymic phases [1]. BD have a critical impact on individuals who suffer from the disease [2], as well as on their family [3,4] and caregivers [5,6]. Family members, often a parent (mother or father) or partner, become the caregiver for adults with BD [4]. Their burden has been described as including impairment in social interactions, financial issues, chronic stress due to fear of relapse or the apprehension of behavioral problems associated with the high and low cycles of BD, and even conflicts within the family [7]. Beyond the examination of the impact of BD on the caregiver, it has also been shown that relationships among family members can also be impacted by BD. Marital relationships are, for instance, marked by difficulties such as volatility or divorces [8,9]. In fact, addressing family functioning has been recognized as being important in the treatment of mental illness in any given individual [10].

In return, it has been shown that family functioning can affect the person with BD [11,12,13]. Negative affective interactions characterized by hostility or emotional over-involvement have been shown to increase the risk of relapse and to have a deleterious effect on the psychosocial functioning of adults with BD [11]. The distress of individuals with a BD in response to these interactions was associated with more severe depressive and manic symptoms [12]. Furthermore, it has been reported that the higher the caregiver’s perceived burden and emotional over-involvement, the worse the adherence of the person with BD to their medication, increasing the risk of a major episode [13]. Taken together, these findings underscore how the intricate relationships between family members may affect the course of BD. To date, scientific literature considers either the impact of BD on family or the impact of family on BD, which is a fragmented view of family functioning. Indeed, in the dynamic system paradigm, family functioning takes into account the interdependence between all family members [14], as well as the affective and behavioral interactions between them [15].

Current data underlines the relevance of assessing family functioning as part of routine clinical practice [7], while several international clinical guidelines for BD recommend the involvement of families in care and propose implementing therapeutic family interventions [16,17,18]. Nevertheless, recommendations do not describe procedures to guide clinical practitioners in the assessment of family functioning. To the best of our knowledge, to date, there is no family functioning assessment battery as there is to assess the individual functioning (cognitive, for example) of a person with BD. The objective of the present systematic review is to contribute to the process of developing such guidelines by examining how family functioning has been assessed when one family member has BD. We specifically investigate the assessment of family functioning in terms of instruments, informants, and family characteristics.

## 2. Materials and Methods

This systematic review was conducted according to the PRISMA (preferred reporting items for systematic reviews and meta-analyses) statement [19] and was not preregistered.

### 2.1. Eligibility Criteria

To be included in this review, articles were required to meet the following eligibility criteria (see Appendix A): articles should (1) be written in English or French; and (2) be an original published research study—review articles, books, book chapters, theses, editorials, guidelines, conference abstracts, indexes and model proposals were excluded; (3) studies should pertain to adults (over 18 years old) diagnosed with BD; and (4) the study should report the assessment of family functioning of the current family and not of the family-of-origin. Indeed, distinct theoretical, methodological, and clinical implications are associated with the assessment of the family-of-origin. Furthermore, articles assessing specific family relationships, such as marital or parental relationships, rather than assessing the whole family functioning were excluded (e.g., [9]). Indeed, while these relationships can be a part of family functioning, family functioning cannot be reduced to them. These concepts are distinct and comprise specific dimensions. The last search was run on 15 January 2020. No publication date restriction was applied.

### 2.2. Information Sources

The search was conducted using the following terms: “family functioning” AND “bipolar disorder” in seven electronic databases: Cochrane (1992–present), PsycArticles (1894–present), Psychology and Behavioral Sciences Collection (1965–present), PsycInfo (1800–present), PubMed (1950–present), ScienceDirect (1823–present) and Scopus (1970–present). The electronic search strategy for each database is presented in Appendix A.

### 2.3. Study Selection

The selection of studies reviewed was performed by three of the authors. After removing duplicates, the first step consisted of screening on the basis of title and abstract. A second screening was performed, searching the full text of the publications to retain those meeting the eligibility criteria. Articles that did not meet the eligibility criteria were excluded.

### 2.4. Data Collection Process

Systematic data extraction from the included studies was carried out and is reported in Table 1. We extracted the data describing the study population characteristics (characteristics of the person with BD and those of other family members), the study objectives, outcomes, instruments used to assess family functioning, informant characteristics, the definition of the family, and study results. The extraction was performed by the first author and was verified by three other authors, to limit the risk of bias.

## 3. Results

The study selection is detailed in Figure 1. In short, a total of 1129 records were identified through the initial search. After removing duplicates, 948 records remained. Each record was then screened for eligibility, and 33 articles met the criteria for inclusion. Among these, only 28 described unique samples, and 5 articles were based on the same sample, focusing on different aspects of the study. Therefore, we retained only one article: the one reporting the highest number of elements regarding the extracted data of the review. Consequently, 29 unique studies were included in this systematic review. The articles were all published in English from 1986 to 2019.

### 3.1. Instruments to Assess Family Functioning

We first focused on the instruments used to assess family functioning (see Table 2). Almost all included studies, 27 (about 93%), used a self-report questionnaire to assess family functioning, and 23 (about 79%) used this method only. The most frequently used questionnaire was the family assessment device (FAD) [20], used in 15 studies (about 52% of the included studies). The FAD, elaborated in the context of the McMaster model of family functioning (MMFF) [21], considers family functioning in terms of seven dimensions: problem-solving, communication, roles, affective responsiveness, affective involvement, behavior control, and general functioning. However, the FAD was sometimes not used in its entirety [22,23]: four studies (about 14% of the studies) only used the general functioning scale [24,25,26,27]. As reported in these studies, the FAD permits dichotomizing family functioning into healthy and unhealthy functioning [24,25,27,28,29,30,31].

The family adaptability and cohesion evaluation scale (FACES) [32] was used in four studies (about 14% of the studies). The FACES is a self-report assessment designed to assess family cohesion and family adaptability, which are the two central dimensions of the Circumplex model of marital and family systems [33]. When coupled with the family communication scale and the family satisfaction scale (FACES IV Package) [34], it assesses communication, a third dimension, as well as satisfaction with cohesion, flexibility, and communication. It is possible to distinguish functional from dysfunctional general family functioning with this questionnaire [35].

The FAD and the FACES were sometimes complemented with other instruments. One study coupled the FAD with a visual analog scale measuring family functioning [29]. The construction of this scale was, however, not reported. In addition to the FACES IV package, another study [35] used the family questionnaire to measure expressed emotion of the key caregiver, the family burden scale to assess the family burden, and the general health questionnaire to evaluate the caregivers’ psychological distress of the key caregiver. The latter instruments allow for the identification of high and low levels of the dimensions they assess. Lastly, the FACES II was also coupled with the conflict behavior questionnaire to assess parent–child communication and conflict [36].

Other self-report questionnaires were also used. For example, Freed et al. [37] and Reinares et al. [38] used the family environment scale. This scale allows a dichotomization of family functioning into positive or negative scores, or a categorization of the cohesion, expressiveness, and conflict areas into low and high levels. To assess family member distress and relationship distress between the person with BD and their family member(s) (parent, partner, sibling, or other), Mueser et al. [39] respectively used the brief symptom inventory and the family attitude scale. The family disruption items of the overall caregiver burden scale [40] and the ICE expressive family functioning questionnaire [41] were also used. The systemic clinical outcome and routine evaluation assessed family adjustment from the perspectives of family strengths, family difficulties, and family communication [42]. Family functioning was also assessed using the confusion, hubbub, and order scale [43] and the family attitude inventory [44].

Six out of 29 studies (about 21%) included a semi-structured interview to assess family functioning. Two studies (about 7%) assessed family functioning solely with this method. Weintraub [45] used the family evaluation form and a semi-structured interview. The second study used a semi-structured interview, focused on wellness, family, and friends [46]. Four other studies (about 13%) coupled a self-report questionnaire with a clinician-rated scale; they performed a multi-method assessment. Indeed, the FAD can be completed with the McMaster clinical rating scale (MCRS), which is also based on the McMaster model (MMFF) [21]; it assesses the same dimensions of family functioning [26]. It is possible to use a cut-off with the FAD to identify unhealthy and healthy functioning [25,31]. The family experiences interview schedule was also used for measuring the perceived burden of the key relative and their gratification, according to their relationship with the person with BD [39].

Lastly, the operationalization of family functioning was not only based on the theoretical model of the instrument; the Calgary family intervention model (CFIM) [47] was also referenced [41]. Suresky et al. [40] studied the family by referring to the resilience theory. Robinson [24] relied on the work of several teams of researchers who had previously published work in the field (McCubbin, Olson, or Epstein’s teams), and Lau et al. [48] rooted their conception of family functioning in the global systemic approach. In summary, all models referenced comprised the systemic approach.

**Table 2 jcm-11-00841-t002:** Instruments to Assess Family Functioning.

	Family Assessment Device (FAD)	Family Adaptability and Cohesion Evaluation Scale (FACES)	Family Environment Scale (FES)	Another Questionnaire	Semi-Structured Interview
Dimensions of Family Functioning	Problem-Solving, Communication, Roles, Affective Responsiveness, Affective Involvement, Behavior Control, and General Functioning	Cohesion and Adaptability	Relationships, Personal Growth, and System Maintenance	No Standardization	No Standardization
[22] Cohen et al. (2013)	x †				
[23] Du Rocher Schudlich et al. (2008)	x †				
[24] Robinson (1996)	x †				
[25] Sheets et al. (2010)	x †				McMaster Clinical Rating Scale
[26] Uebelacker et al. (2006)	x †				McMaster Clinical Rating Scale
[27] Weinstock and Miller (2010)	x †				
[28] Friedmann et al. (1997)	x				
[29] Heru and Ryan (2004)	x				
[30] Miller et al. (1986)	x				
[31] Weinstock et al. (2006)	x				McMaster Clinical Rating Scale
[35] Koutra et al. (2016)		x ‡		Family Questionnaire, Family Burden Scale, and General Health Questionnaire–28-item version	
[36] Shalev et al. (2019)		x		Conflict Behavior Questionnaire	
[37] Freed et al. (2015)			x		
[38] Reinares et al. (2016)			x		
[39] Mueser et al. (2009)				Brief Symptom Inventory, and Family Attitude Scale	Family Experiences Interview Schedule
[40] Suresky et al. (2014)				Overall Caregiver Burden Scale	
[41] Sveinbjarnardottir et al. (2013)				ICE Expressive Family Functioning Questionnaire	
[42] Fitzhenry et al. (2015)				Systemic Clinical Outcome and Routine Evaluation	
[43] Jones et al. (2017)				Confusion, Hubbub and Order Scale	
[44] Clarkin et al. (1990)				Family Attitude Inventory	
[45] Weintraub (1987)					Family Evaluation Form
[46] Wang and Henning (2012)					x
[48] Lau et al. (2018)		x †			
[49] MacPherson et al. (2018)	x				
[50] Berutti et al. (2016)	x				
[51] Koyama et al. (2004)	x				
[52] Müller et al. (2019)	x				
[53] Park et al. (2015)		x			
[54] Weinstock et al. (2013)	x				
TOTAL (% of use of the instrument among studies)	51.7%	13.8%	6.9%	27.6%	20.7%

Note: x the study used the instrument; † not all the dimensions are considered; ‡ FACES IV package, also comprising family communication and family satisfaction scales.

### 3.2. Informant(s)

We found that both the nature and the number of informants were highly variable (see Table 3). In 41% of the studies, only one family member was an informant: either the person with BD [22,26,27,37,42,43,46,49] or another family member, such as the caregiver/key relative [29,35], a female family member [40], or another unspecified adult [41]. Some studies assessed family functioning separately, using the person with BD and the caregiver/key relative [38,50,51] or the partner [25,45] as informants. The perception of family functioning was compared between the person with BD and their family member [38,50,51] or not compared [25,39,45]. In some studies, several family members were used as informants, with the person with BD most often considered among them. In some instances, all family members over the age of 12 years old were used separately as informants and a single average score of family functioning was computed [28,30]. Miller et al. [30] used the average score with and without the score of the person with BD in the analyses. Weinstock et al. [31] and Weinstock et al. [54] used the person with BD and at least one family member separately as informants, without comparing their family functioning scores (they computed a single mean score of the different family members if there was more than one of them). Robinson [24] assessed family functioning using the mother, the father, the sibling, and the person with BD as informants and investigated correlations between their scores. Clarkin et al. [44] reported using the family of the person with BD as informants, without other information. When the entire family was used as informants; most often, semi-structured interviews were used [25,26,31]. Lastly, when the offspring of the person with BD was considered as an informant, family functioning was studied regarding their parents or their caregivers (not necessarily the parent of the caregiver with BD) [23,52,53]. The offspring’s perception of family functioning was also considered and assessed separately from the parents’ perception [36,48].

### 3.3. Family Characteristics

Family characteristics referred to the characteristics of both the adult with BD and their family members. Regarding adults with BD, participants were mostly women (59.7%) with an average age of 38.41 (SD = 6.39) (see Table 4). In 72% of studies, individuals with BD were familiar with the mental health care system: they were outpatients [22,25,31,50,51,54], inpatients [25,27,28,29,30,31,35,37,41,45,54], undergoing partial hospitalization [25,27,31,54] or were inpatients or outpatients at least once during their childhood [52]. Furthermore, some individuals were involved in a therapeutic intervention such as a family-focused treatment [24] or a family-focused and/or pharmacological treatment [25,26,27,31]. Studies included higher proportions of individuals with bipolar I (from 60.8% to 93.8%) compared to individuals with bipolar II disorder (from 6.2% to 20%) [38,43,48], as well as individuals with BD that is not otherwise specified (32.6% of individuals in one study) [49]. Some studies focused on bipolar I only [25,26,27,31,50,54], and two studies included bipolar I and II disorders, but they did not report the percentage of persons within each typology of the disorder [36,53]. 

BD was also characterized by its episode type, which varied across the studies retained. All or most individuals with BD were in the euthymic phase [35,38,49,50], whereas in other studies they were in the acute phase [26,28,29,41], mostly during a manic episode [25,27,30,31,44,54]. BD comorbidities were only reported and taken into account in one-quarter (24%) of the studies [25,30,31,38,39,49,54]. There was a great variability across studies in how comorbidities were managed. For instance, individuals with substance abuse or dependence were often excluded from the sample [25,26,29,31,43,50]. The heterogeneity in BD clinical expression was considered in one study that compared individuals who had a history of suicide attempts to those who did not [50]. Lastly, the retained studies had specific inclusion criteria mentioning the family and thus included individuals with BD who had to live with either a relative or the family [38,50], were in contact with them [22,39], or both [27,31,54]. The family status of the person with BD (i.e., father, mother, child, sibling) was provided in 38% of the included studies [23,24,28,30,36,37,43,45,48,52,53]. 

Having information on family characteristics is important to understand the family that is being assessed. However, apart from the person with BD, the information provided about other family members concerned mostly those who were informants and did not concern the broader family of the person (see Table 4). Some data was provided concerning family structure, such as the number of family members living at home, the number of children, or the intactness of the family. The family of the person with BD was most often intact (i.e., original family), or consisted of single-parent families [24,30,37,43,48]. Two out of 29 articles (about 7%) provided a definition of the family [35,48]. The definition of a family that was presented emphasized the consideration of the family as a whole, with its members interconnected according to a systemic approach.

### 3.4. Family Functioning and Bipolar Disorders

Family functioning characteristics that were obtained specifically in the context of adult BD were derived from twenty articles out of the 29 originally retained. In the other nine articles, results were presented for all the disorders included in the study, and results concerning family functioning, specifically in the context of adult BD, were thus not readily extractable [22,24,35,37,39,41,42,45,52].

Firstly, numerous studies highlighted that BD clinical presentations were associated with family functioning [23,25,26,36]. For example, the greater the depressive symptomatology, the poorer the family functioning (manic and personality disorders symptoms were variables that held statistically constant) [25]. Conversely, studies showed the impact of family functioning on BD: the poorer the family functioning, the poorer the BD course (more manic symptoms and more suicide attempts [50], a longer duration of illness, worse psychosocial functioning, or more hospitalizations [38]). Furthermore, the quality of their relationships within the family was an important aspect in the recovery from BD [46]. Nevertheless, family functioning did not appear as a predictor either of manic or depressive symptoms at a one-year follow-up, when adjusting for immediate post-treatment symptoms [27].

Results comparing BD and other psychiatric disorders were rather inconsistent. Many studies reported that family functioning in individuals with BD was poor, compared to the healthy controls [28,36,38,43,49,53]. Only two studies reported no significant differences between family functioning in individuals with BD and family functioning in healthy controls [30,48]. Heru and Ryan [29] reported that the family functioning assessed in BD was less adapted in comparison to major depression. Other studies showed there were no differences concerning family functioning between the BD group and other psychiatric disorders, such as major depressive disorder and schizophrenia [36,51]. As per Suresky et al. [40], family functioning, as assessed through family disruption, appeared less adapted in BD than in schizophrenia but it was more adapted in major depression (in terms of communication and general functioning) [30]. Such an inconsistency across studies between BD and other psychiatric disorders may reflect variability in both instruments and informants, or that individuals with BD were assessed during different phases of the disorder [31,40].

Family functioning was also studied among different BD groups. Among bipolar I disorder individuals, family functioning was significantly less adapted in those with a history of suicide attempts compared to those without such a history [50]. Family functioning was also compared between adults and youths with BD (7 to 17 years old), demonstrating less adapted family functioning in adults [49]. Descriptively, studies also showed that the score of family functioning fell at or within an “unhealthy” range during an acute episode of BD [31,54], while it was close to the “unhealthy” threshold when individuals were in euthymic episodes [31]. From acute to euthymic episodes, the latter study reported a significant overall improvement of family functioning but less well-adapted family-rated communication (when adjusting for axis I comorbidity and therapeutic or treatment intervention). 

Lastly, studies showed similar perceptions of family functioning between the person with BD and other family members. The person with BD’s perception of family functioning was positively correlated with the perception of other family members [38,50,51]. When both informants’ perceptions were compared, no consistent and significant differences were found [38,51]. It is important to note that in one study [54], this concordance was only found if family members under the age of eighteen were excluded (the perception of family functioning in individuals with BD and the perception of the child/adolescent family members were different).

### 3.5. Risk of Bias in the Assessment of Family Functioning

Regarding the risk of bias, the family functioning assessment was limited to self-reports only in the case of 23 out of 29 studies (almost 79%). Self-report measures are not objective and incorporate biases, such as social desirability [55] and recall bias [56]. A multi-method assessment may reduce these biases. A clinician-based rating may also be added for a more objective evaluation. Nevertheless, only four studies (about 14% of the studies) coupled self-report assessments with a semi-structured interview [25,26,31,39]. The psychometric properties of instruments (i.e, validity and reliability) [57] must be verified. These psychometric properties were variably reported across studies and sometimes no information was given [22,26,29,42,43,44,45,46,53]. One of the biggest limitations derived from the observation was that the instruments for assessing family functioning were not used in their entirety. Most studies considered only a subset of the dimensions originally described in the conceptual models referenced, or combined outcomes from different dimensions to generate a single score. Thus, studies did not apply a systemic methodology, by not considering all the interrelated dimensions of family functioning, even though the family functioning assessment was conceptually attached to the systemic perspective. In addition, there was no consensus across studies regarding the assessment of family functioning. In certain studies, some of the outcomes assessed were defined as dimensions of family functioning. In other studies, these same outcomes were not considered as dimensions of family functioning (e.g., family conflict [23,44]; parent–offspring relationships [48], relationship distress between the patient and their partner [25], or children’s perception of the family environment [45]).

Information regarding the nature of informants was not explicitly reported in many studies. Information regarding which family member was used as an informant was unclear in several studies [25,28,30,31,44]. It was particularly unclear whether the BD parent was evaluated in those studies where offspring were considered [48,52,53]. Moreover, in almost half of the studies, only one person was rated to assess the whole family’s functioning (the person with BD or another family member). To reflect family dynamics, assessing the whole family requires using several informants.

Sufficient information concerning family characteristics was not provided. Firstly, the characteristics of the person with BD were often incomplete. The majority of the studies (17 out of 29—almost 59%) did not characterize the BD type (bipolar I disorder, bipolar II disorder, or not otherwise specified [22,23,24,28,29,30,35,37,39,40,41,42,44,45,46,51,52]. Moreover, one-quarter of the studies did not provide any information on how the subjects managed BD comorbidities, despite BD being characterized by a high rate of comorbidities [58,59]. Furthermore, only 20% of the studies provided complete socio-demographic and clinical data for the studied population (i.e., age, gender, patient status, type of bipolar disorder, episode of bipolar disorder, status in the family). In one-quarter of the studies (eight out of 29), these data were available for the overall sample and not for mental illness subgroups [22,23,24,35,39,41,42], or were combined with data regarding the other family members [30]. Next, the family status of the person with BD was still insufficiently provided (in only eleven studies—almost 38%). However, even if the characteristics of those individuals with a BD were complete or almost complete, their inherent specificities were not investigated (e.g., comparison between individuals in a manic episode and those in a depressive episode), despite the great heterogeneity of BD [60]. Thus, differences in family functioning were not studied according to clinical expressions of BD. Next, the socio-demographic and clinical characteristics of the other family members were heterogeneous across studies. In eight studies (among 27.5%), data was mentioned but not readily extractable because it was presented for all the disorders included in the study [23,24,29,35,39,40,41,54]. More than half of the studies did not provide any information [25,26,28,29,31,36,38,39,40,41,45,50,51,52,53,54]. Information about the family structure was only available in a minority of studies [24,30,37,43,48]. Lastly, for six of the included studies (about 21%), no information about family characteristics was provided [22,27,42,44,46,49]. The lack of pertinent descriptors of the family underscores the need for guidelines by which to study such parameters in the family functioning assessment of an adult with BD.

### 3.6. Risk of Bias in the Current Review

The present review summarizes the findings from a limited number of articles. Importantly, a single team was responsible for seven of the original studies included in the review [25,26,27,28,30,31,54]. Thus, team-specific methodological choices are likely to have affected our review of the scientific literature. Furthermore, our focus on the concept of family functioning may have limited our sampling by excluding studies that addressed family interactions, family conflict, or even family environment (e.g., [61]).

## 4. Discussion

The purpose of the present review is to examine how family functioning has been assessed in the case of families who have a member with BD. It is a particularly relevant topic since the need to consider family functioning in BD management is increasingly recognized [6,7,8,11,13]. Current guidelines recommend the involvement of families in clinical practice [16,17,18]. However, the study of family functioning in the context of BD is still in its infancy. This systematic review finds 29 papers and investigates the assessment of family functioning. Assessing family functioning involves several steps: the choice of instruments, the choice of informants, and the choice of family characteristics are all significant. The main result of the review is that there is heterogeneity in family functioning assessment in terms of the dimensions of family functioning examined and the assessment methods, as well as the instruments used and the family characteristics collected. Proposals for improving the assessment process of family functioning are discussed, as well as the clinical and research implications.

### 4.1. Assessing Family Functioning as a Multidimensional Concept

This review highlights how instruments varied across the studies, although two instruments were frequently used. Firstly, the family assessment device (FAD) [20] was used in half of the studies. Those dimensions of family functioning assessed with the FAD include problem-solving, communication, roles, affective responsiveness, affective involvement, behavior control, and general functioning. Secondly, the family adaptability and cohesion evaluation scale (FACES) [32] was used in 52% of the studies reviewed (15 out of 29 studies). Those dimensions of family functioning assessed with the FACES model are cohesion and adaptability. Several other dimensions of family functioning were assessed using other instruments, including family expressiveness [37,38], family conflict [36,37,38], family distress [39], family disruption [40], or family strengths [42]. Moreover, not all instruments were represented, such as the family APGAR index, assessing family satisfaction with family functioning through five dimensions (i.e., adaptation, partnership, growth, affection, and resolve [62]). This is an illustration of the limitation of this review, due to the exclusion of Spanish as a review criterion [63] and the Google Scholar database [64]. Family functioning is a multidimensional concept. Not all tools propose to assess the same dimensions, depending on the underlying theoretical framework. In the studies, family functioning was often reduced to only one [24,25,26,27,40,48] or a limited set of dimensions [23]. Consequently, a fragmented view of family functioning was considered. Studies assessing several dimensions did not analyze them independently but instead as a total score [22,42,43]. This does not take into account the fact that a part of family functioning or a single total score may not adequately capture the multidimensional nature of family functioning [15,20,33]. In clinical practice, every dimension should be considered to understand family dynamics and to account for the heterogeneity of family functioning. One family functioning parameter is not equal to another regarding the association of its dimensions. As is consistent with a personalized medicine approach [65,66], identifying the heterogeneity of family functioning is important to propose an adapted treatment according to each family functioning scenario. To do so, a complementary approach, called a person-oriented approach [67], could be used by identifying subpopulations with the same association between dimensions of family functioning, and thus with the same family functioning scenario. Mean scores on the dimensions of the entire population are no longer performed but are rather associations between individuals having similar scores regarding these dimensions (as does clustering) [67,68,69]. A plurality of family functioning can be highlighted, with high scores on some dimensions and low scores on others. Therefore, in clinical practice, this means that practitioners can work on the weaknesses of a family while building on the family’s strengths. However, to date, no study has proposed such a dynamic vision of family functioning when one family member has BD.

### 4.2. Performing a Multi-Method Assessment of Family Functioning

The great majority of studies used self-reporting to assess family functioning, and two studies relied on a clinician-rated measurement of family functioning [45,46]. Studies focused on the point of view of the person or that of the clinician. Both are relevant but both have biases, such as social desirability [55] and recall bias [56], or as heuristics of representativeness and availability [70,71]. Subjective perceptions have a crucial role in BD [72,73], but the use of crossed assessment is relevant in clinical practice and research, particularly in order to have a more holistic view of family functioning. For instance, a crossed assessment will allow practitioners to reduce the gap between the person’s perception of family functioning and the clinician’s evaluation of family functioning. In a more collaborative conception, it will also allow for the discussion of discrepancies or similarities between the person’s perception and that of the clinician, to make shared decisions about care. Therefore, in both clinical practice and scientific studies, multi-method assessments combining self-reporting with interviews may hold advantages [25,26,31,39], as is recommended by the guidelines for the assessment of symptomatology [74].

### 4.3. Using Several Family Members as Informants

The number and nature of informants varied greatly across the studies. Twelve out of 29 studies (about 41%) relied on a single informant, either the person with BD or another family member, such as the partner, the mother, or the father [26,27,29,35,37,40,41,42,43,46,49]. Some studies have included multiple informants, including the person with BD, and have compared their scores [24,38,50,51], or they have considered an average trend within the family by computing a mean score of all family members’ scores (the person with BD sometimes being included) [28,30,31,54]. For future research, it could be innovative to assess the different perceptions of family members since family functioning is the result of the associations of the family members’ perceptions [75]. Thus, it would be relevant to propose a model of family interactions. In line with this conception, analyses such as dyadic analysis [76], using an actor-partner interdependence model [77], or social network analysis [78] should be performed. For instance, such analyses can account for the dynamic interaction between individuals by testing the interdependent relationship between two family members, or by investigating the network of interdependent relationships within the family. In research, it appears that the family functioning assessment could thus be improved by multiplying the informants and by using analyses that take into account the perception of each informant. In clinical practice, evaluating the family members’ various perceptions may allow clinicians to achieve a view of family functioning as close as possible to the reality co-constructed through the perceptions of each family member. Consequently, it will be possible to propose specifically adapted, personalized care to the whole family. 

### 4.4. Characterizing the Family 

The selection of family members used as informants is crucial to have the best overview of family functioning. Defining the family and understanding the characteristics of family configuration is relevant. On the one hand, family functioning may be quite different according to the family member’s clinical expression of BD. Due to the great heterogeneity of BD [60], one BD expression is not equal to another. This concept is emphasized by the results of certain included studies showing that family functioning in BD was different according to suicidality [50], the phase (acute or recovery) [31,54], or even the age [49]. Providing systematically socio-demographic and clinical characteristics of individuals with BD will allow us to grasp the diversity of BD and allow us to assess family functioning according to clinical expressions of BD. On the other hand, apart from the person with BD, the characteristics of the other family members and information about family structure are needed. The included studies, however, lacked the reporting of characteristics, particularly in terms of family structure: only five studies provided information [24,30,37,43,48] such as the intactness of the family. However, specific family interactions can differ according to family structure, like those of adolescents [79]. Thus, broader family functioning may also differ. Studies also lacked the reporting of characteristics of family members other than the informants. We understand that family functioning may differ according to the socio-demographic and clinical characteristics of the member with BD, but it can also vary according to those of the other family members. Consequently, assessing family functioning, taking into account the heterogeneity of the clinical expression of BD, and studying the characteristics of other family members will allow better representativeness and transferability of the results regarding current practice, in line with personalized medicine [65,66]. To assist researchers, a consensual grid of the socio-demographic and clinical characteristics of family members, including the person with BD, should be proposed. For example, a systematic assessment could be used to demonstrate the total number of family members, the number of members at home, the status within the family of each family member (e.g., mother, father, grandfather, daughter, partner), the intactness of the family, the duration of the marital relationship, or even the total number of children and their age.

### 4.5. Family Functioning as a Determinant of Remission: The Need for Assessment Guidelines

Overall, the results of this systematic review show that, to date, there is no consensual assessment of family functioning, although the latter is a crucial axis of care. In fact, the results of the included studies showed that individuals with BD are characterized by less adapted family functioning compared to those healthy individuals [28,36,38,43,49,53] that impact on and are impacted by the disorder. Family functioning is associated with psychosocial functioning and symptomatology severity in BD [27,38,46,50]. Family functioning is, thus, associated with the person’s remission [74]. In turn, remission has implications on the family: both symptomatology and functional impairment in adults with BD affect family functioning [25,26,36]. Family functioning and remission are, thus, interrelated. Family functioning might be a crucial factor to investigate in practice, knowing the severe impact of symptomatology and functional impairment on the individuals’ daily lives [80,81,82]. Family functioning is an important variable to assess but there is yet to be a gold standard instrument by which to assess it. The lack of standardization in assessment limits the construction of guidelines for assessing family functioning in BD. In the pursuit of a personalized medicine approach [65,66], the assessment of family functioning should emphasize the heterogeneity of family functioning and the clinical expression of BD. To do so, it must rely on multiple methods of assessment, multiple informants, and a description of family members. 

## 5. Conclusions: Toward Personalized Care, a Systemic Person-Oriented Approach

Studying family functioning in the context of bipolar disorder is complex, as the interaction between their family’s specificities and the person’s clinical expression of BD is at play [83]. To complement the assessment of family functioning in BD, we further support a multidimensional approach to identify both family functioning and the clinical expression of BD profiles from a systemic person-oriented approach [84,85]. Such an approach should better investigate how a profile of BD expression can be associated with a profile of family functioning. Assessing such profiles may significantly benefit clinicians in the care of adults with bipolar disorder and their families. 

## Figures and Tables

**Figure 1 jcm-11-00841-f001:**
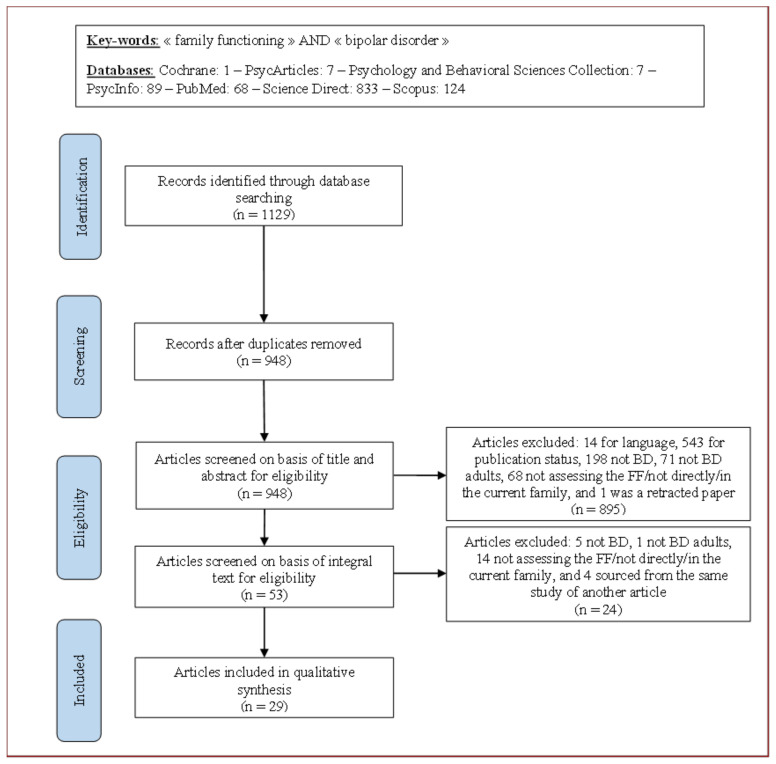
Flowchart of the study selection. Note: BD = Bipolar disorders; FF = Family functioning.

**Table 1 jcm-11-00841-t001:** Data extraction.

**Population**	Country	Socio-Demographic and Clinical Characteristics of People with BD ^1^	Socio-Demographic and Clinical Characteristics of Other Family Members	Family Structure
**Objective(s)**	As reported in the main text or in the abstract			
**Outcome(s)**	Outcomes considered to assess FF ^2^			
**Instrument(s)**	Instrument assessing FF ^2^	Validation	Specified if cut-off is considered	
**Informant(s)**	Who is the informant for FF? ^2^			
**Underlying theoretical framework**	Definition of family	Is the assessment of FF ^2^ based on a theoretical framework?		
**Results**	Relevant results concerning FF ^2^ in BD ^1^ specifically			

^1^ BD = Bipolar disorders; ^2^ FF = Family functioning.

**Table 3 jcm-11-00841-t003:** Informants.

	Number of Informants		Informant Status		
	One	Several	Person with a BD	One Family Member of the Person BD	Several Family Members of the Person with BD
[22] Cohen et al. (2013)	x		x		
[23] Du Rocher Schudlich et al. (2008)	x		x	x	
[24] Robinson (1996)		x	x		x
[25] Sheets et al. (2010)		x	x		x
[26] Uebelacker et al. (2006)		x	x		x
[27] Weinstock and Miller (2010)	x		x		
[28] Friedmann et al. (1997)		x	x		x
[29] Heru and Ryan (2004)	x			x	
[30] Miller et al. (1986)		x	x		x
[31] Weinstock et al. (2006)		x	x		x
[35] Koutra et al. (2016)	x			x	
[36] Shalev et al. (2019)		x	x		x
[37] Freed et al. (2015)	x		x		
[38] Reinares et al. (2016)		x	x	x	
[39] Mueser et al. (2009)		x	x	x	
[40] Suresky et al. (2014)	x			x	
[41] Sveinbjarnardottir et al. (2013)	x			x	
[42] Fitzhenry et al. (2015)	x		x		
[43] Jones et al. (2017)	x		x		
[44] Clarkin et al. (1990)		x †	x †		x †
[45] Weintraub (1987)		x	x	x	
[46] Wang and Henning (2012)	x		x		
[48] Lau et al. (2018)		x	x		x
[49] MacPherson et al. (2018)	x		x		
[50] Berutti et al. (2016)		x	x	x	
[51] Koyama et al. (2004)		x	x	x	
[52] Müller et al. (2019)		x	x		x
[53] Park et al. (2015)		x	x		x
[54] Weinstock et al. (2013)		x	x		x
TOTAL	41.4%	58.6%	86.2%	34.5%	41.4%

Note: x the study used the instrument; † unclear information; BD: Bipolar Disorder.

**Table 4 jcm-11-00841-t004:** Characteristics of the family members (person with BD and their family members).

	**Country**	**Total Sample**	Persons with BD Characteristics							Family Members’ Characteristics of the Person with a BD	
			Sample Size	BD Type	Patient Status	BD Period	Sex (% of Females)	Age Mean (SD)	Status in Family	Sample Size	Status in Family
[22] Cohen et al. (2013)	USA	104	104	NI	Outpatient	NI	Confused	Confused	NI	Not included	
[23] Du Rocher Schudlich et al. (2008)	USA	231	77	NI	NI	NI	Confused	Confused	Parent	154	Offspring with mental illness
[24] Robinson (1996)	USA	NI	7	NI	NI	NI	Confused	Confused	Child	Confused	Parent Sibling
[25] Sheets et al. (2010)	USA	112	56	Type I	Inpatient Outpatient Partial	Acute Manic (82%)	55%	41.77 (None)	NI	56	Partner
[26] Uebelacker et al. (2006)	USA	62	62	Type I	NI	Acute	58%	40.6 (12.5)	NI	None	Partner (63%) Parent (21%) Adult child/another adult (16%)
[27] Weinstock and Miller (2010)	USA	92	92	Type I	Inpatient Outpatient Partial	Manic (75%) Depressive (20%) Mixed (5%)	57%	39 (11.5)	NI	Not included	
[28] Friedmann et al. (1997)	USA	171	60	NI	Inpatient	Acute	77%	38.2 (12.9)	Partner (74%) Child (26%)	111	Over the age of 12
[29] Heru and Ryan (2004)	USA	10	Non included	NI	Inpatient	Any	NI	Confused	NI	10	Confused
[30] Miller et al. (1986)	USA	30	15	NI	Inpatient	Manic	Confused	Confused	Partner (80%) Child (20%)	15	Over the age of 12
[31] Weinstock et al. (2006)	USA	NI	69	Type I	Inpatient Outpatient Partial	Manic	59.4%	38.80 (11.03)	NI	Average of 1.54 per patient	Partner (71%) Parent (21.7%) Child/other (2.9%)
[35] Koutra et al. (2016)	Greece	18	18	NI	Inpatient	Euthymic	Confused	Confused	NI	18	Confused
[36] Shalev et al. (2019)	USA	737	256	Type I Type II	NI	NI	NI	NI	Parent	481	Offspring
[37] Freed et al. (2015)	USA	192	75	NI	Inpatient	NI	68%	43.62 (6.77)	Parent	117	Offspring
[38] Reinares et al. (2016)	Spain	164	82	Type I (83%) Type II (17%)	NI	Euthymic	51%	34.67 (10.0)	NI	82	Partner (40%) Parent (54%) Adult child/Sibling (6%)
[39] Mueser et al. (2009)	USA	58	29	NI	NI	NI	Confused	Confused	NI	29	Confused
[40] Suresky et al. (2014)	USA	33	Non included	NI	NI	NI	NI	NI	NI	33	Woman family member
[41] Sveinbjarnardottir et al. (2013)	Iceland	68	34	NI	Inpatient	Acute	Confused	Confused	NI	34	Confused
[42] Fitzhenry et al. (2015)	Ireland	18	18	NI	Confused	NI	Confused	Confused	Confused	Not included	
[43] Jones et al. (2017)	UK	97	97	Type I (94%) Type II (6%)	NI	NI	78%	36.65 (None)	Parent	Not included	
[44] Clarkin et al. (1990)	USA	21	21	NI	Inpatient	Manic (61.9%) Depressive (33.3%) Mixed (4.8%)	67%	32.3 (15.4)	NI	Not included	
[45] Weintraub (1987)	USA	250	58	NI	Inpatient	NI	44.83%	NI	Parent	58 134	Partner Child
[46] Wang and Henning (2012)	New Zealand	9	9	NI	NI	NI	55.56%	41 (None)	NI	Not included	
[48] Lau et al. (2018)	Australia	149	59	Type I (79.6%) Type II (20.3%)	NI	NI	NI	NI	Parent	90	Offspring
[49] MacPherson et al. (2018)	USA	46	46	Type I (60.9%) Type II (6.5%) Nos (32.6%)	NI	Euthymic (80%)	61%	21.16 (2.77)	NI	Not included	
[50] Berutti et al. (2016)	Brazil	113	62	Type I	Outpatient	Euthymic	60.9%	41 (None)	NI	51	Partner Parent
[51] Koyama et al. (2004)	Japan	36	18	NI	Outpatient	NI	44.4%	49.4 (10.5)	NI	18	Partner (94.4%) Parent (5.6%)
[52] Müller et al. (2019)	Denmark	NI	NI	NI	NI	NI	NI	NI	Parent	NI	Partner Child
[53] Park et al. (2015)	USA	64	NI	Type I Type II	NI	NI	NI	NI	Parent	64	Offspring
[54] Weinstock et al. (2013)	USA	227	92	Type I	Inpatient Outpatient Partial	Manic (75%) Depressive (20%) Mixed (5%)	57%	39.57 (11.30)	NI	135	Partner (67%) At least 1 parent (20%) At least 1 sibling (3%) At least 1 other (10%)

Note: confused = confused in the total sample; NI = non informed; nos = bipolar disorder not otherwise specified.

## Data Availability

This a PRISMA review, data extracted from studies are available in the main text and summarized in the tables.

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
