# Peer review of "Improving the Assessment Process of Family Functioning in Adult Bipolar Disorders: A PRISMA Systematic Review"

_jcm, 2022, doi:10.3390/jcm11030841_

Round 1

Reviewer 1 Report

The present study aimed to systematic review how family functioning is assessed in adult bipolar disorders which is a critical area of investigation. I have a few questions/comments:

Q1: line 79, in eligibility criteria, why the study exclude the assessment of the family-of -origin? As we know, the functioning of family-of-origin is also very important for the family members, family structure, family atmosphere and also psychopoahology of bipolar disorder.

Q2: line 118, in 3.1. Instruments to assess family functioning. The measures of family functioning were well reviewed and summarized in the present study. But the Family APGAR Index was another useful assessment to measure the family functioning, we did not see it in the study.

Q3: line 159. “Six out of 29 studies (about 21%) included a semi-structured interview to assess family functioning. Two studies assessed family functioning solely with this method. Weintraub (1987) used the Family Evaluation Form, a semi-structured interview. The second study used a semi-structured interview focused on wellness, family and friends[46].”  To best knowledge, the study (Granek,L.; Danan,D.; Bersudsky, K.; Osher, K. Living with Bipolar Disorder: The Impact on Patients, Spouses, and Their Marital Relationship. Bipolar Disorders 2016, 18, doi:10.1111/bdi.12370.) also used a semi-structured interview focused on marital relations in bipolar disorder.  

Q4: line 373. “Furthermore, only 20% of the studies provided complete socio-demographic and clinical data for the studies population.” The complete socio-demographic and clinical data refer to ??

Q5: line 487. characterizing the family. “It is also important to better understand what family type family functioning is characteristic of.” The family type is another interesting topic. It will be better if the present study could describe the assessments of family type, such as balance type etc.   

Author Response

Responses to the referee Number 2 

This manuscript is a systematic literature review of research that compares family functioning in bipolar disorder (BD). It finds that the literature is limited and suggests how research could be improved.

ABSTRACT

Because the numbers of articles are small (29 in total), the results section of the Abstract should include numbers of articles as well as %, i.e. say “27 of 29 articles (93%)” etc.  The results section of the Abstract could be expanded slightly to make the data clearer.

Response

  • We would like to thank reviewer 2 for his/her guidance. We thus added numbers of articles in the Abstract section: “27 out of 29 studies (93%) relied on self-report questionnaires, 12 out of 29 studies (41%) used one family member as an informant”. However, because of the limited number of words in the abstract section, we unfortunately cannot expand the results section of the Abstract.

INTRODUCTION

This is clear and self-explanatory.

METHODS

This is clear and concise. However Table 1 is a bit confusing at first glance and it might be better to put the headings down the Y axis rather than across the top?

Response

  • For better clarity, we reverse the axis of the table 1 a suggested by reviewer 2:

 Table 1. Data Extraction

Population

Country

Socio-demographic and clinical characteristics of people with BD1

Socio-demographic and clinical characteristics of other family members

Family structure

Objective(s)

As reported in the main text or in the abstract

Outcome(s)

Outcomes considered to assess FF2

Instrument(s)

Instrument assessing FF2

Validation

Specified if cut-off is considered

Informant(s)

Who is the informant for FF?2

Underlying theoretical framework

Definition of family

Does the assessment of FF2 is based on a theoretical framework?

Results

Relevant results concerning FF2 in BD1specifically

1 BD = Bipolar Disorders; 2FF = Family Functioning.

RESULTS

Need to explain what excluding 543 articles for “publication status” means.

Response

  • Articles excluded for publication status were review articles, books, book chapters, theses, editorials, guidelines, conference abstracts, indexes and model proposals. Indeed, we only included original research studies in this review. We clarified this information by added in the 1. Eligibility criteria in the Methods section the terms “publication status” : (2) referring to the publication status, be a published original research study; review articles, books, book chapters, theses, editorials, guidelines, conference abstracts, indexes and model proposals were excluded”

When reporting of the 29 studies and various subsets of that small group, use numbers as well as percentages each time as this is clearer to the reader.

Response

  • We reviewed the article by reporting systematically the numbers as well as the associated percentage of the article :

“The most frequently used questionnaire was the Family Assessment Device (FAD)[20], used in fifteen studies (about 52% of the included studies)”

“four studies (about 14% of the studies) only used the general functioning scale [24–27]”

“The Family Adaptability and Cohesion Evaluation Scale (FACES)[32] was used in four studies (about 14% of the studies).”

“The family status of the person with BD (i.e., father, mother, child, sibling) was provided in eleven studies (about 38%)”

“Family functioning characteristics obtained, specifically in the context of adult BD, were derived from twenty articles out of the 29 retained originally (about 70%). In the nine other articles (about 30%), results were presented for all the disorders included in the study.”

“Nevertheless, only four studies (about 14% of the studies)  coupled self-report assessments with a semi-structured interview [25,26,31,39].”

“Indeed, the majority of the studies (seventeen out of 29 - almost 598%) did not characterize the BD type”

“In one fourth of the studies (seven out of 29), these data were available for the overall sample and not for mental illness subgroups [22–24,35,39,41,42];”

“Lastly, the family status of the person with BD was still too little provided (in only eleven studies - almost 38% of the included studies)”

“In eight studies (among 27.5%), data was mentioned but not readily extractable because it was presented for all the disorders included in the study [23,24,29,35,39–41,54].”

“Lastly, for six included studies (about 21%), no information about family characteristics was provided [22,27,42,44,46,49].”

“Secondly, the Family Adaptability and Cohesion Evaluation Scale (FACES)[32] was used in 52% of the 29 of studies reviewed (fifteen out of 29 studies)”

“Twelve out of ten 29 studies (about 41%) relied on a single informant”

Line 152 – remove the “a” for better English grammar.

Response

  • We remove the “a”: “between the person with BD and his/her family member (parent, partner, sibling or other)”

Lines 187-211 – these detailed results should be added to the results section of the Abstract, as the manuscript is long and some readers will focus only on the Abstract.

Response

  • We provided results related to 2 Informant(s) in the Abstract section by summarizing the data: “12 out of 29 studies (41%) used one family member as an informant (adult with BD or other) and the adult considered was mostly a woman in acute phase of bipolar I disorder”. Because of the limited number of words in the abstract section, we unfortunately cannot provide more detailed information.

Lines 239-241 – It is debatable whether to use the terms “overrepresented” and “underrepresented” as some experts argue that BD-II is overdiagnosed or represents milder cases of BD-I.  Perhaps better to say “more” BD-I and “less BD-II.

Response

  • Reviewer 2 drew our attention to an important point. We followed his/her recommendations by writing “more” instead of “overly represented” and “less” instead of “underrepresented” : “Individuals with bipolar I disorder were more represented (from 60.8% to 93.8%) while individuals with bipolar II disorder were less represented (from 6.2% to 20%)”.

Lines 292-293:  Please provide more information on what “(manic and personality disorders symptoms held constant)” means.  It is counter-intuitive to think there was no worsening in family functioning if manic symptoms present versus a euthymic phase of BD?  This statement also seems incompatible with lines 322-327 where family functioning worsened in acute episodes.

Response

  • We apologize for the lack of clarity of our sentence. Indeed, by “manic and personality disorders symptoms held constant” we meant that in the statistical analysis (regression), these symptoms were controlled in order to appreciate the effect of depressive symptomatology only. We now understand that our sentence is misleading, because it suggests that family functioning held constant instead of the presence of manic of personality disorders symptoms which is, as reviewer 2 said, counter-intuitive. We thus clarified our sentence by writing: “(manic and personality disorders symptoms were variables held statistically constant)”.

Lines 334-336 interesting as might reflect that children experience reduction in parenting capacity when parent in acute phase of BD.

Response

  • We thank reviewer 2 for this comment.

Lines 338-339 appear to conflict with the Abstract:  one statement says 80%, the other says 93% of studies relied on self-report.  Either correct or clarify why the difference please.

Response

  • As reported in 3.1. Instruments to assess family functioning of the Results section, 79% of the included studies relied only on self-report, whereas 13% relied on self-report but also on interview. Thanks to the comment, we understand this data might be confusing in all sections. Thus, we clarify in the 5. Risk of bias in the assessment of family functioning of the Results section by adding “only” in the sentence and by reporting the same percentage: “Regarding the risk of bias, family functioning assessment was limited to self-reports only for 23 out of 29 studies (almost 79%)”.

Lines 390-391 are a good summary of the main message of this systematic review: “The lack of 389 pertinent descriptors of the family underscores the need for guidelines to study such 390 parameters in the family functioning assessment of an adult with BD.”

Response

  • We thank reviewer 2 for this comment.

DISCUSSION

Lines 432-434, for grammatical correctness and clarity of meaning please consider adding the words in bold, or something similar:

“A family functioning parameter is not equal to another 432 regarding the association of its dimensions. Consistent with a personalized medicine approach 433 [62,63], identifying the heterogeneity of family functioning is important to propose an 434 adapted treatment according to each family functioning.”

Response

  • We are grateful to reviewer 2 for his/her help with the writing in English. We added the terms proposed by reviewer 2: “A family functioning parameter is not equal to another, regarding the association of its dimensions. Consistent with a personalized medicine approach [62,63]”.

Line 449 – typo the word “one” should be “on”.

Response

  • We are grateful to reviewer 2 for the writing corrections. We corrected the sentence: “Studies are focused either on the point of view of the person”.

Line 452 – should the word “of” be added where I’ve put it in bold?:  “So, subjective perceptions have a 451 crucial role in BD [69,70], but use of crossed assessment is relevant in clinical practice and in 452 research, particularly...”

Response

  • We agree with reviewer 2 and we corrected the sentence: “So, subjective perceptions have a crucial role in BD [69,70], but the use of crossed assessment is relevant in clinical practice and in research”.

Line 508 – For better English grammar don’t have “a” in this sentence: “Indeed, consistent with a personalized medicine...”  and the whole sentence would benefit from restructuring for better grammar and clarity: “Indeed, consistent with a personalized medicine [62,63], it will allow to know what family 508 type family functioning is characteristic of, and consequently to know if results 509 correspond to the type of family we meet in clinical practice .”

Response

  • We removed the “a” and rephrased/restructured the sentence: “Indeed, consistent with personalized medicine [62,63], it will allow to know what family type family functioning is characteristic of. Thus, it will allow to know if empirical evidence correspond to the type of family we meet in clinical practice.”

Line 525-527 – This sentence needs improved grammar to provide more clarity:  “Symptomatic and functional remissions 525 constitute a crucial health issue which is important to improve regarding the considerable 526 interrelated impact of the symptomatology and the functional impairment on the 527 individuals’ daily lives [77–79].”

Response

  • According to the reviewer 2 suggestion, we rephrased the sentence and restructured the paragraph : “Family functioning is thus associated to the person’s remission [71]. In turn, remission has implications upon the family: both symptomatology and functional impairment in people with BD affect the family functioning [25,26,36]. Family functioning and remission are thus interrelated. Family functioning might be a crucial factor to investigate in practice knowing the severe impact of the symptomatology and the functional impairment on the individuals’ daily lives [77–79].”

Line 528 – more grammatical to say “remission has implications upon the family”

Response

  • We are grateful to reviewer 2 for the writing corrections. We corrected the sentence: “Family functioning is thus associated to the person’s remission [71]. In turn, remission has implications upon the family”.

Line 536 – again slight adjustment in grammar for flow:  “To do so, it must rely on a multiple methods of assessment, on a 536 multiple informants, and on a description of family members.”

Response

  • We are grateful to reviewer 2 for the writing corrections. We corrected the sentence: “To do so, it must rely on several methods of assessment, on multiple informants, and on description of family members.”

Lines 543 to 561 – Having read the manuscript I can follow what the authors mean in this conclusion, however the language used is somewhat technical and some readers might skim the article and focus on the conclusion – it might be better to use some plainer language for the same meaning in this section.

Response

  • To allow readers to better understand the meaning of this section, we rephrased the paragraph: “Considering a single summary indicator such as a mean score for health or family functioning status allows a better understanding of the study population and a better estimation of its behavior. However, this approach does not account for the variability of the clinical and family functioning expressions. Thus, to complement the current assessment of family functioning in BD, we support a multidimensional approach of both family functioning and BD clinical expression. Thus, associations between dimensions form qualitatively distinct profiles [85]. Such an approach should better investigate how a profile of BD expression can be associated with a profile of family functioning. Profiles of family functioning may be in adaptation to those of the clinical expression of BD as well as profiles of the clinical expression of BD may be in adaptation to family functioning profiles.”

FINAL COMMENT

The manuscript has involved a lot of work and shown how research in this area is important and needs to be improved upon, it suggests how those improvements can be made.  The Results and Discussion are lengthy and there are some grammatical improvements that could be made, I have not necessarily found or pointed out all of them – it might help to have a review by an English language expert to see if it can be shortened using grammatical changes that don’t remove content or meaning.

Response

Reviewer 2 Report

The authors have focused on and examined family functioning for the treatment of bipolar disorder, which is unique and would be a necessary point of view. However, since this is a broad theme, they should focus more on a certain point, make hypotheses and examine this. There are no hypotheses or conclusions but only negative data and results in their manuscript. The authors need to rewrite their manuscript as a whole.

Author Response

Responses to the referee Number 1 

To address the comments of reviewer 1, we have emphasized the value of this literature review article in bringing family into the care and research. Indeed, there is a lack of pertinent descriptors of the family, so it underscores the need for guidelines to study such parameters in the family functioning assessment of an adult with BD. In this way, the present systematic review does not have hypothesis but have an aim: to investigate how family functioning is assessed in the context of bipolar disorders. According to the comments, we have rewritten a part of the results and the conclusion firstly to better emphasize research on the domain, and secondly to better highlight our analysis of the scientific literature. This systematic review found 29 papers and investigated the assessment of family functioning. Data shows how research in this field is important and needs to be improved upon to better consider family both in care and research. This suggests how those improvements can be made. Assessing family is not about the choice of the instrument only. Indeed, assessing family is a process in several steps: the choice of instrument, the choice of informants and the choice of family characteristics.

Reviewer 3 Report

Peer review of jcm-1470739. Nov.2021

This manuscript is a systematic literature review of research that compares family functioning in bipolar disorder (BD). It finds that the literature is limited and suggests how research could be improved.

ABSTRACT

Because the numbers of articles are small (29 in total), the results section of the Abstract should include numbers of articles as well as %, i.e. say “27 of 29 articles (93%)” etc.  The results section of the Abstract could be expanded slightly to make the data clearer.

INTRODUCTION

This is clear and self-explanatory.

METHODS

This is clear and concise. However Table 1 is a bit confusing at first glance and it might be better to put the headings down the Y axis rather than across the top?

RESULTS

Need to explain what excluding 543 articles for “publication status” means.

When reporting of the 29 studies and various subsets of that small group, use numbers as well as percentages each time as this is clearer to the reader.

Line 152 – remove the “a” for better English grammar.

Lines 187-211 – these detailed results should be added to the results section of the Abstract, as the manuscript is long and some readers will focus only on the Abstract.

Lines 239-241 – It is debatable whether to use the terms “overrepresented” and “underrepresented” as some experts argue that BD-II is overdiagnosed or represents milder cases of BD-I.  Perhaps better to say “more” BD-I and “less BD-II.

Lines 292-293:  Please provide more information on what “(manic and personality disorders symptoms held constant)” means.  It is counter-intuitive to think there was no worsening in family functioning if manic symptoms present versus a euthymic phase of BD?  This statement also seems incompatible with lines 322-327 where family functioning worsened in acute episodes.

Lines 334-336 interesting as might reflect that children experience reduction in parenting capacity when parent in acute phase of BD.

Lines 338-339 appear to conflict with the Abstract:  one statement says 80%, the other says 93% of studies relied on self-report.  Either correct or clarify why the difference please.

Lines 390-391 are a good summary of the main message of this systematic review: “The lack of 389 pertinent descriptors of the family underscores the need for guidelines to study such 390 parameters in the family functioning assessment of an adult with BD.”

DISCUSSION

Lines 432-434, for grammatical correctness and clarity of meaning please consider adding the words in bold, or something similar:

“A family functioning parameter is not equal to another 432 regarding the association of its dimensions. Consistent with a personalized medicine approach 433 [62,63], identifying the heterogeneity of family functioning is important to propose an 434 adapted treatment according to each family functioning.”

Line 449 – typo the word “one” should be “on”.

Line 452 – should the word “of” be added where I’ve put it in bold?:  “So, subjective perceptions have a 451 crucial role in BD [69,70], but use of crossed assessment is relevant in clinical practice and in 452 research, particularly...”

Line 508 – For better English grammar don’t have “a” in this sentence: “Indeed, consistent with a personalized medicine...”  and the whole sentence would benefit from restructuring for better grammar and clarity: “Indeed, consistent with a personalized medicine [62,63], it will allow to know what family 508 type family functioning is characteristic of, and consequently to know if results 509 correspond to the type of family we meet in clinical practice .”

Line 525-527 – This sentence needs improved grammar to provide more clarity:  “Symptomatic and functional remissions 525 constitute a crucial health issue which is important to improve regarding the considerable 526 interrelated impact of the symptomatology and the functional impairment on the 527 individuals’ daily lives [77–79].”

Line 528 – more grammatical to say “remission has implications upon the family”

Line 536 – again slight adjustment in grammar for flow:  “To do so, it must rely on a multiple methods of assessment, on a 536 multiple informants, and on a description of family members.”

Lines 543 to 561 – Having read the manuscript I can follow what the authors mean in this conclusion, however the language used is somewhat technical and some readers might skim the article and focus on the conclusion – it might be better to use some plainer language for the same meaning in this section.

FINAL COMMENT

The manuscript has involved a lot of work and shown how research in this area is important and needs to be improved upon, it suggests how those improvements can be made.  The Results and Discussion are lengthy and there are some grammatical improvements that could be made, I have not necessarily found or pointed out all of them – it might help to have a review by an English language expert to see if it can be shortened using grammatical changes that don’t remove content or meaning.

Author Response

Reviewer #3:

The present study aimed to systematic review how family functioning is assessed in adult bipolar disorders which is a critical area of investigation. I have a few questions/comments:

Q1: line 79, in eligibility criteria, why the study exclude the assessment of the family-of -origin? As we know, the functioning of family-of-origin is also very important for the family members, family structure, family atmosphere and also psychopathology of bipolar disorder.

Response:

  • We agree with the Reviewer, family-of-origin has an important role in family dynamics and bipolar disorders. However, focusing on the current family of the person with bipolar disorder does not have the same theoretical, methodological, and clinical implications. On the one hand, with the deinstitutionalization of patients, adults with bipolar disorder spend more time in their current family implying that family members can deal with objective and subjective damages (such as impairment in social relationships, financial issues, chronic stress) but also with an impact in their family relationships (g., Granek L, Danan D, Bersudsky Y, Osher Y. Living with bipolar disorder: the impact on patients, spouses, and their marital relationship. Bipolar Disord. 2016;18(2):192-199. doi:10.1111/bdi.12370). On the other hand, family functioning (e.g., a negative affective climate) has an impact on the course of the disorder in terms of symptomatology and psychosocial functioning (Miklowitz DJ. Family Factors and the Course of Bipolar Affective Disorder. Arch Gen Psychiatry. 1988;45(3):225. doi:10.1001/archpsyc.1988.01800270033004). Thus, assessing family functioning of the current family has clinical implications for the person having the disorder as well as for the other family members, currently living together, currently interacting together. It will allow us to work on the current family interactions to improve the family members’ health (including the person with bipolar disorder). For us, this clinical objective is crucial regarding the burden, but it is still too little considered in practice.
  • However, the focus on the family-of-origin functioning involves a developmental perspective with the person’s construction or even the biological and environmental factors. Clinical implications could be to work with the person having the disorder by considering his/her perceptions of the functioning or even his/her psychological patterns (schemas) built on the family-of-origin’s interactions. It might be less beneficial to the current family interactions and thus to the other family members. Concerning the instrument used, items might not be the same because they do not refer to the same underlying concept.
  • Thus, specific distinct implications are related to the assessment of the family functioning of the current family and to the functioning of the family-of-origin. Consequently, we support that is important to make the distinction between these two assessments. For all these reasons, we chose to focus our systematic review on the family functioning of the current family.
  • Finally, it is important to precise this non-eligibility criteria had a very low impact on our synthesis of the literature since only one article was excluded for assessing functioning of the family-of-origin.
  • According to the relevance of the remark of the reviewer 3, we have added in the 2.1 Eligibility criteria the justification: “(4) report the assessment of family functioning of the current family and not of the family-of-origin. Indeed, specific distinct implications (theoretical, methodological, and clinical) are related to the assessment of the family functioning of the current family and to the functioning assessment of the family-of-origin. This justifies the relevance of two distinct reviews. This review focuses on the interactions of the current family with regard of the clinical issues for all the family members (including the person with BD) as it is mentioned in the Introduction part.”

Q2: line 118, in 3.1. Instruments to assess family functioning. The measures of family functioning were well reviewed and summarized in the present study. But the Family APGAR Index was another useful assessment to measure the family functioning, we did not see it in the study.

Response:

  • We are grateful to reviewer 3 for this interesting remark. The Family APGAR Index did not appear in our study because none of the included studies used this instrument.
  • The Family APGAR Index assesses family satisfaction with family functioning through five dimensions (i.e., Adaptation, Partnership, Growth, Affection, Resolve - APGAR) which could be very relevant in bipolar disorders both in research and clinical practice. Indeed, by clustering family functioning in three degrees (highly dysfunctional, moderately dysfunctional, highly functional), this measure might allow us to adapt the care according to the profile consistent with personalized medicine as we supported in our study.
  • Thanks to the remark, we found two articles (e., In English Doornbos (1996) and Pardo et al. (2011) in Spanish) on the Google Scholar database. These articles were not identified on the seven databases used in the review. This highlights the need in our field to make accessible more scientific journals in order to improve the knowledge about family functioning in bipolar disorders.

Q3: line 159. “Six out of 29 studies (about 21%) included a semi-structured interview to assess family functioning. Two studies assessed family functioning solely with this method. Weintraub (1987) used the Family Evaluation Form, a semi-structured interview. The second study used a semi-structured interview focused on wellness, family and friends[46].”  To best knowledge, the study (Granek,L.; Danan,D.; Bersudsky, K.; Osher, K. Living with Bipolar Disorder: The Impact on Patients, Spouses, and Their Marital Relationship. Bipolar Disorders 2016, 18, doi:10.1111/bdi.12370.) also used a semi-structured interview focused on marital relations in bipolar disorder.  

Response:

  • We thank reviewer 3 for raising the debate about the definition of family functioning. The study of Granek and colleagues (2016) was not included in the review precisely because it focused on marital relationships. While marital relationships can be part of family functioning (as well as parental relationships for example), family functioning cannot be reduced to marital relationships. Both are two distinct concepts comprising specific dimensions, and thus instruments used for the assessment are different as well as the objective of the assessment. Likewise, in clinical practice, treating marital relationships is not the same care that treating the whole family interactions. Overall, that is why we tried to emphasize in our study how important it is to define family functioning and, in the best case, to be based on a theoretical model.

Q4: line 373. “Furthermore, only 20% of the studies provided complete socio-demographic and clinical data for the studies population.” The complete socio-demographic and clinical data refer to ??

Response:

  • This sentence was not clear indeed. We referred particularly to the data mentioned in 3 Family characteristics such as age, gender, patient status, type of bipolar disorder, episode of bipolar disorder, status in family. Thanks to the remark, we have added this clarification: “only 20% of the studies provided complete socio-demographic and clinical data for the studied population (i.e., age, gender, patient status, type of bipolar disorder, episode of bipolar disorder, status in family)”

Q5: line 487. characterizing the family. “It is also important to better understand what family type family functioning is characteristic of.” The family type is another interesting topic. It will be better if the present study could describe the assessments of family type, such as balance type etc.   

Response:

  • We are grateful to reviewer 3 for his/her very relevant remark. Indeed, this remark allows us to rethink what we meant through family type. As reviewer 3 pointed out, family type referred to a specific conception that we did not address in this part. By family type, we meant here the configuration of the family according to the family members’ characteristics and the structure of family. Thus, we have changed the word “type” by “configuration” to be more precise: “It is also important to better understand what family configuration family functioning is characteristic of.”
  • Secondly, it would be very interesting if we could have described the assessments of family type, but it would be quite difficult, and it does not address our objective. Indeed, this remark emphasizes precisely one of the results of this review: scientific literature did not assess the family functioning in order to bring out types of functioning, but to characterize the functioning in terms of an average score. However, a possibility for us was to take the scores found in the studies, and characterize the family functioning according to the family type built with these mean scores. But it was not the purpose of this review: the objective was to highlight how the assessment is currently done and what we can have as results with such an assessment.